# HALLUCINATIONS IN NEURAL MACHINE TRANSLATION

## ABSTRACT

Neural machine translation (NMT) systems have reached state of the art performance in translating text and are in wide deployment. Yet little is understood about how these systems function or break. Here we show that NMT systems are susceptible to producing highly pathological translations that are completely untethered from the source material, which we term *hallucinations*. Such pathological translations are problematic because they are are deeply disturbing of user trust and easy to find with a simple search. We describe a method to generate hallucinations and show that many common variations of the NMT architecture are susceptible to them. We study a variety of approaches to reduce the frequency of hallucinations, including data augmentation, dynamical systems and regularization techniques, showing that data augmentation significantly reduces hallucination frequency. Finally, we analyze networks that produce hallucinations and show that there are signatures in the attention matrix as well as in the hidden states of the decoder.

## 1 INTRODUCTION

Neural machine translation (NMT) systems are language translation systems based on deep learning architectures (Cho et al., 2014a; Bahdanau et al., 2014; Sutskever et al., 2014). In the past few years, NMT has vastly improved and has been deployed in production systems, for example at Google (Wu et al., 2016), Facebook (Gehring et al., 2017), Microsoft (Hassan et al., 2018), and many others. As NMT systems are built on deep learning methodology, they exhibit both the strengths and weaknesses of the approach. For example, NMT systems are competitive with state of the art performance (Bojar et al., 2017) and scale well to very large datasets (Ott et al., 2018) but like most large deep learning systems, NMT systems are poorly understood. For example, in many commercial translation systems, entering repeated words many times occasionally results in strange translations, a phenomenon which has been highly publicized (Christian, 2018). More broadly, recent work shows that NMT systems are highly sensitive to noise in the input tokens (Belinkov & Bisk, 2017) and also susceptible to adversarial inputs (Cheng et al., 2018). When there is an error in translation, it can be challenging to either understand why the mistake occurred or engineer a fix.

Here we continue the study of noise in the input sequence and describe a type of phenomenon that is particularly pernicious, whereby inserting a single additional input token into the source sequence can completely divorce the translation from the input sentence. For example, here is a German input sentence translated to English (reference) by a small NMT system:

```
Source: Caldiero sprach mit E! Nachrichten nach dem hart erkämpften Sieg,
    noch immer unter dem Schock über den Gewinn des Großen Preises
    von 1 Million $.
Reference: Caldiero spoke with E! News after the hard-fought victory, still
    in shock about winning the $1 million grand prize.
NMT Translation: Caldiero spoke with E, after the hard won victory,
    still under the shock of the winning of the Grand Prix of 1 million $.
```

Simply by adding a single input token (*mit:with*, *werden:to become* and *dass:that*) to the beginning of the input sentence and translating them with the same NMT model yield the following:

```
Mistranslations:
    mit: It was said to have a lot of fun in the world.
    werden: Don't hesitate to contact us, if you want to be able to pay for you.
    dass: I'm looking forward to having a lot of money.
```

These mistranslations are completely semantically incorrect and also grammatically viable. They are untethered from the input so we name them *'hallucinations'*. Clearly, even if hallucinations occur only occasionally, the NMT model may lose user trust and/or lead the user to a false sense of confidence in a very incorrect translation. In this work, we show that hallucinations are widespread in the popular NMT system we study. For example, 73% of sentences in our test set can be perturbed to hallucination in our simplified canonical model (with a average, greedily-decoded BLEU score of 21.29). If we decode with beam search (average BLEU 25.66) that number drops, but remains high at 48%.

We systematically explore hallucinations in a variety of NMT variants. We also develop methodologies to help ameliorate this problem, using ideas from data augmentation, dynamical systems theory, and regularization. Finally, we analyze an NMT model that demonstrates hallucinations and show that there are signatures of hallucinations that can be seen in the attention matrix. We focus on NMT systems built using RNNs (Wu et al., 2016), on which many commercial translation systems are based. As RNNs recursively generate translations, we examine their dynamical nature. Finally, we study decoder stability when hallucinations are produced and when they are not.

## 2 RELATED WORK

Since its invention, researchers have been working to better understand NMT. For example, moving from the original Seq2Seq model Sutskever et al. (2014); Cho et al. (2014b) to models that utilize attention mechanisms (Bahdanau et al., 2014), resulted in improved translation quality (Luong et al., 2015) and better interpretability (Ding et al., 2017). Studies identified the most critical components of the LSTM (Greff et al., 2017) and the role of the LSTM in language modeling (Karpathy et al., 2015) more broadly. Followed by explorations in interpretability, recent work has focused on robust NMT, studying the effects of input noise, aiming to reduce variation from typos (Belinkov & Bisk, 2017) and synonym choice (Cheng et al., 2018) in the discrete input set used by NMT systems. Both (Belinkov & Bisk, 2017) and (Cheng et al., 2018) have discovered that NMT systems are highly sensitive to input noise and both used adversarial training to help stabilize NMT systems (either with black-box adversarial training or augmenting an adversarial loss). There has been work in understanding how to handle some of the pathologies in RNNs for language modeling and translation, for example, using scheduled sampling to handle mismatches between training and test sets (Bengio et al., 2015).

In parallel, there has also been work in understanding RNNs through the framework of dynamical systems (Sussillo & Barak, 2013; Sussillo, 2014; Laurent & von Brecht, 2016; Rivkind & Barak, 2017; Beer & Barak, 2018) and understanding their capacity in language tasks (Collins et al., 2017). For example, it is known that continuous time vanilla RNNs exhibit high-dimensional chaos (Sompolinsky et al., 1988), which can be driven out by strong input (Rajan et al., 2010). RNNs exhibiting chaos can be beneficial, for example, RNNs capable of chaos in the absence of input can serve as strong initialization for training RNNs when they are input driven (Sussillo & Abbott, 2009), but caution must be used as unbridled chaos can be viewed as a source of dynamical noise. This has led to efforts to rid RNNs of chaos altogether (Laurent & von Brecht, 2016). Efforts related to improving optimization may also have dynamically regularizing effects, e.g. (Arjovsky et al., 2015). Given the complexities and slow compute times of recurrent systems, there have also been attempts to rid NMT of recurrence (Kalchbrenner et al., 2016; Gehring et al., 2017; Vaswani et al., 2017).

We further expand on these studies by highlighting the specific pathology of hallucinations, systematically studying those hallucinations, and analyzing them from a dynamical systems perspective.

## 3 NMT METHODS

**Models**: In this paper, we use a standard RNN-based encoder-decoder NMT model with attention. Specifically, we study the NMT model described in (Wu et al., 2016), known as GNMT. We use the GNMT model and its public implementation[1]. Formally, given an input sequence, $\mathbf{x}_{1:S}$ of length $S$, the NMT model first encodes the input sequence $\mathbf{x}_{1:S}$ into a set of vectors $\mathbf{z}_{1:S} = f_{enc}(\mathbf{x}_{1:S})$ using its encoder $f_{enc}$. The task of the decoder $f_{dec}$, is to generate the translation, $\mathbf{y}_{1:T}$, one symbol at a time $y_i$, given the encoding, $\mathbf{z}_{1:S}$, and previously generated symbols $y_{<i}$. The decoder, $f_{dec}$ is implemented as a conditional sequence model (Bahdanau et al., 2014), where the distribution over $\mathbf{y}_{1:T}$ is conditioned on $\mathbf{x}_{1:S}$. The decoder internally

---

[1] https://github.com/tensorflow/nmt with *gnmt-v2* architecture.

makes use of an attention mechanism $f_{att}$ to query the encoder, summarizing $\mathbf{z}_{1:S}$ for each output symbol $y_i$, putting it all together $y_i = f_{dec}(y_{<i}, f_{att}(\mathbf{z}_{1:S}))$ (also see Figure 7 for a detailed model schematic). Finally, the conditional probability of the target sequence is modelled as $p(\mathbf{y}_{1:T}|\mathbf{x}_{1:S}) = \prod_{i=1}^{T} p(y_i|y_{<i}, \mathbf{x}_{1:S})$ and the log of this conditional likelihood is maximized given a set of source-target pairs $(\mathbf{x}, \mathbf{y})$ during training.

We study models that are significantly smaller and less complex than those typically used in state-of-the-art or production systems for the sake of research tractability. We use a single layer bidirectional LSTM in the encoder $f_{enc}$ and a two layered unidirectional LSTM in the decoder $f_{dec}$ with an additive attention mechanism as $f_{att}$ (Britz et al., 2017). The word embedding dimensions and each LSTM hidden cell (both in the encoder and decoder) are set to 256. We refer to this model as the *canonical* model. Unless otherwise stated, we used Adam (Kingma & Ba, 2014) optimizer with a learning rate of 0.001, a constant learning rate schedule, and clipped gradients to a maximum norm of 5.0 during training.

Given these hyper-parameter and architectural choices, we trained 10 canonical models with different random seeds to observe how parameter initialization variability played a role in our results. All additional model variants we describe later were also trained 10 times with the same 10 different random seeds. Each model was trained for 1M steps (updates) with a mini-batch of size 128 and the training checkpoint with the best BLEU score on the development set was selected.

The central goal of our study was to understand how various modelling choices affected the frequency of hallucinations. In order to isolate the effects of modeling changes, all model variants we study in this paper were identical to the canonical model except for a single change. This means, for example, that our model with 512 hidden units also is 2 layers deep, etc. We performed a simple hyper-parameter search for the canonical model, and did not perform additional hyper-parameter searches for any additional models. All models we present are well trained with a BLEU score of at least 20.0 on the test set using greedy decoding, a reasonable score for 2-layer models with 256 hidden units. With beam search decoding, our canonical models achieve an average BLEU score of 25.66.

**Inference**: Generating a translation of the input sequence, or formally finding an output sequence that maximizes the conditional log-probability, $\hat{\mathbf{y}} = \text{argmax}_{\mathbf{y}} \log p(\mathbf{y}|\mathbf{x})$, is a major challenge in NMT since the exact inference (or decoding) is intractable. NMT uses approximate decoding techniques which we also have used in this paper. The simplest approximate decoding technique, *greedy decoding*, chooses the most-likely symbol under the conditional probability $\hat{y}_t = \text{argmax}_i \log p(y_t = i|\hat{y}_{<i}, \mathbf{x}_{1:S})$, outputting a single best local prediction by keeping track of a single hypothesis $k$, at each time step. Another approximate decoding technique, *beam search*, improves upon greedy decoding by keeping track of multiple hypotheses (*beams*), where $k > 1$ at each time step of the decoding, compared to $k$=1 in greedy-decoding. To maintain simplicity in our canonical model we used greedy decoding. Note that production systems will often perform beam search to find a more probable translation than one generated by greedy search. We also ran an additional set of experiments with beam search.

**Data**: We trained all models with the German to English WMT De→En 2016 dataset (4,500,966 examples) (Bojar et al., 2016), validated with the WMT De→En 2015 development set (2,169 examples). We then used the WMT De→En 2016 test set (2,999 examples) to compute the hallucination percentage for each model.

For the input and output of all NMT models in consideration, we used sub-word tokens extracted by Byte-Pair Encoding (BPE) (Sennrich et al., 2015). To construct a vocabulary of unique tokens, we first combined the tokenized source and target corpora[2], and then learned a joint BPE code with an 8k merge operations budget, resulting in 12,564 unique tokens. Further, in order to study the effect of larger vocabulary sizes, for some experiments we repeated the same process with 16,000 and 32,000 BPE codes and ended up with vocabularies having 19,708 and 36,548 unique tokens respectively. Note that we used the same vocabulary for both source and target side languages.

---

[2]We used Moses tokenizer: https://github.com/moses-smt/mosesdecoder/blob/master/scripts/tokenizer/tokenizer.perl

---

**Algorithm 1:** Computing the percentage of hallucinations in a NMT model

---

Select a model; Fix a random seed;
Select a group of subword tokens with the following attributes:;
- Common tokens: 100 most common subword tokens;
- Mid-frequency tokens: random sample of 100 subword tokens between common and rare tokens;
- Rare tokens: 100 least common subword tokens;
- Punctuation: all punctuation tokens;
**for** *every sentence in test corpus (e.g. WMT De→En 2016 test set)* **do**
    **if** *adjusted BLEU between reference sentence and translated sentence* $> 0.09$ **then**
        **for** *every selected token* **do**
            **for** *every location in (beginning, end, second-to-end, randomly in the middle)* **do**
                put the selected token at the selected location in the byte-pair encoded input sequence;
                translate the perturbed input sequence;
                **if** *adjusted BLEU between the translated, perturbed sentence and the translated,*
                  *unperturbed sentence* $< 0.01$ **then**
                    this sentence can be perturbed to produce a hallucination;

---

## 4  HALLUCINATIONS

Informally, a hallucination is a translation of a perturbed input sentence that has almost no words in common with the translation of the unperturbed sentence. Here, we use the term *perturb* to mean adding a single token to the input source sequence. This is based on an initial observation that adding a rare token to an input sequence reliably caused a model to generate a hallucination, e.g. adding a Chinese character token to a German to English translation. We expanded and systematized this discovery into a brute force search procedure (Algorithm 1) by splitting our tokens into several types: common (100 most common German tokens), rare (100 least common German tokens), mid-frequency tokens (randomly sampled 100 tokens from the remaining German tokens), and punctuation tokens. Additionally, we attempted to perturb each sentence by inserting a token at one of several positions: beginning, end, second to the end, and randomly in the middle. We did this for every sentence in our test set and collected statistics for each model variant.

To define a quantitative threshold for a hallucination we modified the BLEU score, which is used to compare a reference sequence with a translated sequence. Briefly, the BLEU score is a common metric for translation, which measures weighted n-gram overlaps while penalizing short translations. We modified the BLEU score by re-weighting the n-grams in the BLEU score computation to favor having any words in common between the two sentences ($1.0$ for one-grams and $0.8$ for bi-grams and disregarded other n-grams). Then, we call only sentences that have an adjusted BLEU score of less than $0.01$ hallucinations. For examples of sentence pairs with different adjusted BLEU scores, see Section 8.2 in the Appendix.

Not all translations are good even before adding a perturbation token to the input sequence. To strengthen our results on hallucinations, we first excluded these poor translations by computing the adjusted BLEU score between the reference translation and the translation produced by the unperturbed input sequence. We kept sentences that had an adjusted BLEU of $\geq 0.09$. We chose a value of $0.09$ because it seemed to maintain enough context that you could tell the translation and the reference were related.

## 5  HALLUCINATION FREQUENCY AND REDUCTION

### 5.1  EXAMPLES

We describe four common hallucination patterns: (1) grammatically correct output that bears no relation to the input text, (2) ungrammatical output with oscillatory structure, (3) output that remains largely in the source language, and finally (4) terse jumps to the end of the sequence. We also observe translations that are ungrammatical nonsense. While still highly undesirable, we note that a user should at least be able spot and reject these additional hallucination patterns. See the Appendix section 8.3 for more examples.

**Source**: Freundschaft schließen durch Backen.
**Reference**: Make friends through baking.
**Perturbation**: Added *ich:I* randomly in sentence.
**Grammatically correct hallucination**:Should you want to join us?

**Source**: Monatelang war hinter verschlossenen Türen verhandelt, gegrübelt,
    debattiert und gezeichnet worden.
**Reference**: Plans have been negotiated, mulled over, debated, and plotted
    behind closed doors for months.
**Perturbation**: Added *uns:we* at the beginning of the sentence.
**Oscillatory hallucination**: In the month of the month of the month of the
    month of the month of the month, it was a matter of course.

**Source**: Neue Verhandlungen mit den Piloten
**Reference**: New negotiations with pilots
**Perturbation**: Added *mit:with* randomly in sentence.
**Source language hallucination**: Neuist e mehr Jahren mit der Piloten d

**Source**: Für die Fed-Repräsentanten beeinflussen die Marktturbulenzen die
    komplexe Kalkulation , wann man die Zinsen erhöhen solle.
**Reference**: For Fed policymakers, the market turmoil adds to the complex
    calculus of when to raise the interest rate.
**Perturbation**: Add *uns:we* randomly in sentence.
**End of sequence hallucination**: For FF.C.

## 5.2    FREQUENCY OF HALLUCINATIONS

We show that hallucinations can be easily evoked by inserting tokens in the source sequence. We used Algorithm 1 to quantify how susceptible a given model is to hallucination. In particular, we studied what types of perturbations (location, and token type) are more effective at inducing hallucinations. With this method, we found that, on average, 73% of all sentences in the WMT De→En test set can be perturbed to hallucination in the canonical model.

We studied how beam search, number of hidden units, vocabulary size, and decoding scheme affected hallucination percentages (Figure 1, left). We found that changing the number of hidden units to both 512 and 1024 from 256 and changing the vocabulary size–from 8K to 16K BPE codes did not significantly decrease the hallucination percentage. However, beam search and a vocabulary size increase corresponding to 32K BPE codes did significantly lower the mean percentage of hallucinations. We also studied how different types of perturbations impacted the hallucination percentage of the canonical model (Figure 1, right). By far, adding a perturbing token to the beginning of the input sequence induces the most hallucinations in the canonical model.

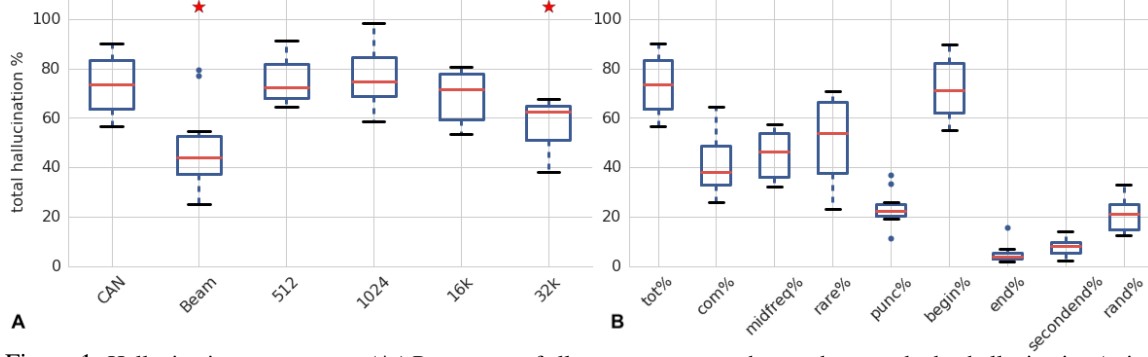

Figure 1: Hallucination percentages. (**A.**) Percentage of all test set sentences that can be perturbed to hallucination (using any type of hallucination token at any location) as a function of model variant: canonical, beam search decoding, number of hidden dimensions (512 and 1024), and vocabulary size (16K and 32K BPE codes). Red stars indicate model variant hallucination % is statistically lower than the hallucination % in the canonical model. (**B.**) Hallucination percentages as a function of perturbation token type and position inserted in the input sentence over all canonical model random seeds. Any particular statistic, e.g. common% is computed over all perturbation locations, likewise begin% averages over all perturbing token types.

We were curious if BLEU scores were predictive of hallucination percentage. We plot the BLEU vs. hallucination percentage of all models we study (Figure 2). Surprisingly, we found that hallucination percentage does not decrease as BLEU score increases. However, since we did not study all possible models, we urge caution in interpreting these results.

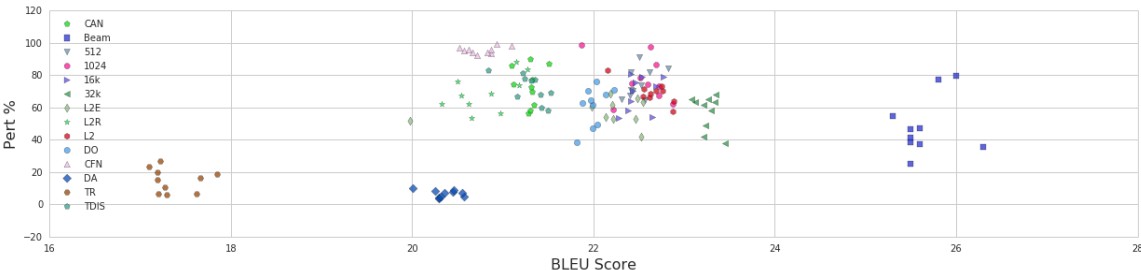

Figure 2: Relationship between BLEU score and hallucination percentage for all models. We do not find hallucination percentage decreases as BLEU score increases. We found a correlation coefficient of .33 between BLEU and hallucination percentage with a significant p-value ($p < 0.001$). While the correlation is significant, we cautiously interpret these results because we do not fully explore the space of all possible models. The Transformer models are the only models we analyzed with $< 20$ BLEU which we explain in section 5.3. For the expansion of acronyms, please also see section 5.3.

## 5.3 REDUCING HALLUCINATION FREQUENCY

What changes can we make to the model to make it more robust to hallucinations? We investigated the effect of three different methodologies, (1) simple regularizations, (2) data augmentation and (3) regularizations on the dynamics in state space. We tested if a model variation significantly reduces hallucinations by performing a one-sided Mann–Whitney U between the canonical distribution of models and the distribution of models that use the model variant. We use a p-value of 0.05.

**Simple Regularizations:** We choose dropout, L2 regularization on embeddings (L2E), L2 regularization on recurrent weights (L2R) and L2 regularization on all weights (L2) as straight-forward regularization techniques to be applied. For dropout, we created a model with dropout in all feed-forward layers, with a keep probability of 0.9. Next, we implemented L2 regularization on the token-embeddings and throughout the entire model with weighting hyperparameters of $1 \times 10^{-4}$ and $1 \times 10^{-5}$, respectively.

**Data augmentation (DA):** We augmented the training data by perturbing all training sentences with a random token (either common, rare, mid-frequency, or punctuation) at either the beginning, end, second-to-end, or randomly in the middle while keeping the reference translation the same. This doubled our training set. We then trained a canonical model with the augmented training set, and found that data augmentation helps decrease hallucination percentages. We call this model DA.

**Dynamical Regularizations:** We wondered if hallucinations could be reduced by providing a more exact initial state for the decoder, so we trained additional models where the initial state of the decoder was tied to last step of the encoder (TDIS). Note that the canonical model sets the decoder initial state as a vector of zeros. As a second regularization method that operates on the state space, we used Chaos-free network (CFN) (Laurent & von Brecht, 2016) which by premise cannot produce chaos. We replaced the LSTM cell with the CFN in a set of experiments, again using 256 hidden units.

Dropout, L2E, and DA all resulted in statistically significant decreases in hallucination percentage, with DA being by far the most effective at decreasing hallucinations. On the contrary, switching out LSTM cells for CFN cells resulted in a significant increase in the hallucination percentage. Finally, linking the initial state of the decoder with the final state of the encoder had no statistical effect on the hallucination percentage.

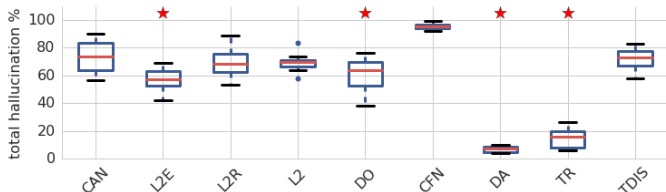

Figure 3: Hallucination percentages for model variants: dropout, L2E, L2R, L2, CFN, DA, TR, and tied decoder initial state (TDIS). The red stars denote a statistically significant difference ($p < 0.05$) between the hallucination % of the model variant and the canonical model.

Although data augmentation dramatically reduced hallucinations in the canonical model, it requires knowing the kind of perturbations that one would use to induce a hallucination. To study how fine grained one's knowledge must be, we trained the canonical model on a different training set where we withheld two types of data augmentation: perturbing at the beginning or with common tokens (We call this model DA w/o beginning or common). We then compared this model with the canonical model trained with the full DA training set (Figure 4). We found that DA w/o beginning or common yields much higher hallucination percentages when tested by perturbing at the beginning or with common tokens in comparison to the DA model. However, we also saw a reduction in hallucination percentage for common and beginning tokens when compared to the canonical model. This indicates that DA can still provide some protection against hallucinations, even if exact perturbations are not known.

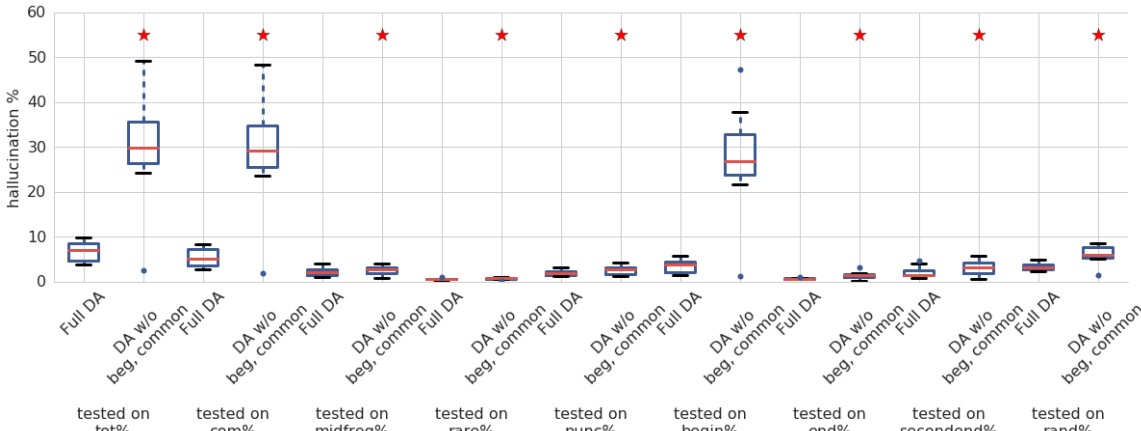

Figure 4: Effects of data augmented (DA) training when including all perturbation types vs excluding common and beginning perturbation types. We trained two models, one including all perturbations types for DA training, and the other excluding common and beginning perturbation types. We then examined the hallucination percentage of each perturbation type for both of these models and studied whether a DA model would be less prone to hallucinate when perturbed with types of tokens or positions it had not been trained against. Red star shows that DA w/o beginning or common had statistically significantly reduced mean compared to the canonical model trained without DA.

Additionally, we wondered if hallucinations were present in NMT architectures that were not recurrent. Thus, we studied the Transformer model (TR) Vaswani et al. (2017). To make our results easily accessible to NMT practitioners, we chose a hyperparameter set from those given in the popular Tensor2Tensor library[3] that was closest to our cutoff BLEU score when using greedy decoding. These models are trained with the parameters from transformer_tiny (2 hidden layers, 128 hidden size, 512 filter size, and 4 heads) and have a greedy BLEU score of 17.5, which is a little lower than our GNMT models. We find the transformer model hallucinates significantly less than the canonical model, but can still be perturbed to hallucinate on average 15% of the time (Figure 3). We present these results with many caveats. Unlike the canonical model, this model is trained with many types of regularization (dropout, attention dropout, label smoothing, relu dropout, and a larger batch size) and a longer input sequence length (256 versus 50 in the canonical model). Unfortunately, training with no regularization or a sequence length of 50 dramatically reduced the BLEU score for parameter combinations we tried, and thus we decided to present these results with caveats instead of a model without regularization and a comparable sequence length.

## 6 ANALYSIS OF HALLUCINATIONS

### 6.1 ATTENTION MATRICES

We observed a large difference between attention matrices of normal translations and of hallucinations. Attention networks in normal translations tend to study the entire input sequence throughout decoding. In French to English and other language pairs that are grammatically aligned (German to English is somewhat aligned), this often results in a strong diagonal in the attention matrix. The attention matrix, when translating hallucinations, however, shows the model attending to a few tokens. We give an example comparison in Figure 5, top panels. For additional attention matrices, see Section 9.

---

[3]https://github.com/tensorflow/tensor2tensor

We wanted to quantify this difference in a way that does not require strong alignment between languages, i.e. one expects English to French to result in a largely diagonal matrix but not English to Turkish, so we used information entropy to compute a statistic that described the difference between attention matrices during a decode that resulted in a hallucination and those that resulted in a normal translation. Specifically, at each output of the decoder, the attention network gives a distribution over input tokens. We averaged these distributions across all decoded output tokens, resulting in a distribution of average attention weight over the input tokens. We treated this as a discrete distribution and computed the entropy, $-\sum_t p(x_t) \log p(x_t)$, where $x_t$ is the input token at time $t$, for each example, resulting in a distribution of entropy values over all decoded sequences.

We then compared the entropy of average attention distributions between hallucinations and correct translations (Figure 5, bottom panels). This figure shows a significant difference between the entropy values for hallucination sequences. As a control, we show there is no significant difference between original input sequences and perturbed input sequences for sentences that cannot be perturbed to hallucination. Note that, in real world scenarios where a ground truth translation is not available, the measure of entropy of the average attention distribution may be useful to detect hallucinations.

The breakdown of the attention module seems to signal that the encoder and the decoder have been decoupled, and the decoder ignores context from the encoder and samples from its language model. Two possibilities are that broken attention modules are the root cause of decoupling, or they are a symptom of further breakdown in the dynamics of the decoder.

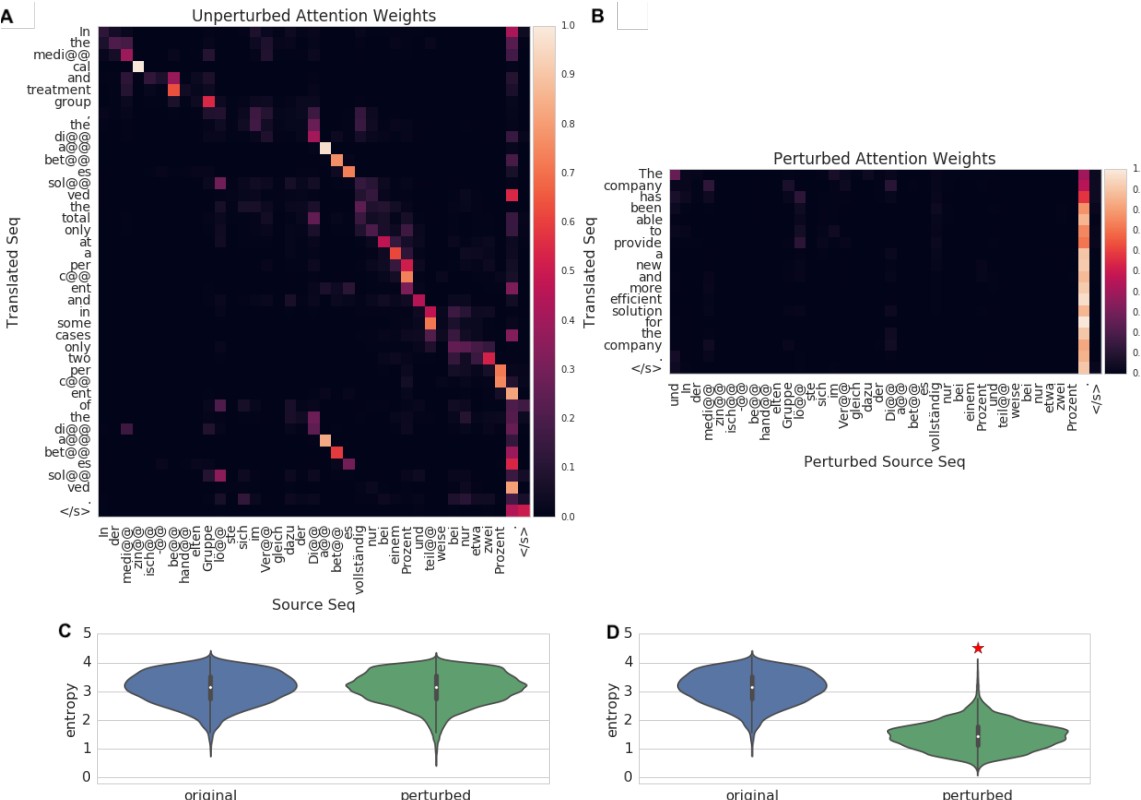

Figure 5: The attention matrix can reveal hallucinations. The attention matrix shows how much weight is applied to each token in the source sequence while decoding each token in the resulting translated sequence. **A.** Normal translations produce attention matrices that show a distribution of weight across most tokens in the source sequence throughout decoding (x-axis source sequence, y-axis decoded sequence). **B.** However, during hallucinations, the attention network tends to place weight on only a few input tokens. We can see the majority of the weight throughout decoding is placed on the "." in the source sequence. **C, D.** We used an entropy measure to quantify how distributed the attention is over the input source sequence. Shown are the distributions for normal translations **C.** and hallucinations **D.** for the original (blue) and perturbed (green) attention entropy values. Mean of the entropy distributions for hallucinations are statistically significantly different (Mann–Whitney U test, $p < 0.05$).

## 6.2 STATISTICS AND STABILITY OF DECODER WHEN PERTURBED

Examining the causes and types of hallucinations: instability of translation, translations decoupled from the input, and oscillations in translations, led us to believe that hallucinations result from a dynamical process gone wrong in the decoder (which might be caused by the encoder or attention module). Further, many model variants that decrease hallucinations (like L2 regularization) can be viewed as regularizing the dynamics of the model. Thus, we explore differences in the decoder between translating a hallucinating or non-hallucinating sentence by comparing the hidden states of the decoder and analyzing the stability of the system (Figure 6).

In this section, we are interested in how perturbations change the decoding pipeline and thus study an unchanged, input sentence (which we call "original" and denote by $\mathbf{x}^o$) and its perturbations ($\mathbf{x}^p$). We perturbed all source sentences in the test set with all types of tokens (common, rare, mid-frequency, and punctuation) at all locations (beginning, end, second-to-end, and randomly) and sorted all perturbations into two groups: those that result in hallucinations and those that do not.

We hypothesize that differences in translation occur early in decoding and focus on the first timestep the decoder receives context from the encoder, $t = 1$. We studied both the distance between (left panel) and ratio of the norms (middle panel) of the perturbed decoder hidden states $h_1(\mathbf{x}^p)$ and the original decode, $h_1(\mathbf{x}^o)$, at decode $t = 1$. Both resulted in obviously different distributions as a function of whether the perturbation resulted in a hallucination. Given these differences (Figure 6), it seemed natural to attempt to causally reduce the number of perturbations by re-normalizing $h_1(\mathbf{x}^p)$ to the norm of $h_1(\mathbf{x}^o)$. We tried this for all perturbations of all original sentences in our test set and did not see a reduction in hallucination percentage.

We wondered if a hallucination results from exciting an unstable mode of the decoder early in the decode. To study this, we defined an unstable subspace of the original decode $U(\mathbf{x}^o)$ as the subspace spanned by the eigenvectors of $\left[ \left( \frac{\partial h_1}{\partial h_0}(\mathbf{x}^o) \right)^T \left( \frac{\partial h_1}{\partial h_0}(\mathbf{x}^o) \right) \right]$ with corresponding eigenvalues greater than 1. We projected the normalized hidden state of the perturbed input, $\mathbf{x}^p$, onto this subspace

$$E(\mathbf{x}^o, \mathbf{x}^p) \equiv \frac{|\hat{h}_1(\mathbf{x}^p)^T U(\mathbf{x}^o)|^2}{|\hat{h}_1(\mathbf{x}^o)^T U(\mathbf{x}^o)|^2}, \tag{1}$$

where $\hat{h}$ is $h$ normalized to 1. We did this for every original sentence in our test set that had at least 10 hallucinations (around half of all original sentences). We show the count (up to 10) of perturbed sentences such that $E(\mathbf{x}^o, \mathbf{x}^p) > 1$ when $\mathbf{x}^p$ resulted in a hallucination (red) and when it did not (blue) in Figure 6 (right panel).

Finally, we also studied the stability exponents of the decoder, focusing on how eigenvalues of the Jacobian, $\frac{\partial h_T}{\partial h_0}(\mathbf{x})$, of the hidden states of the decoder changed as a function of whether or not a perturbation resulted in a hallucination for models trained with and without data augmentation (Shown in Appendix 10).

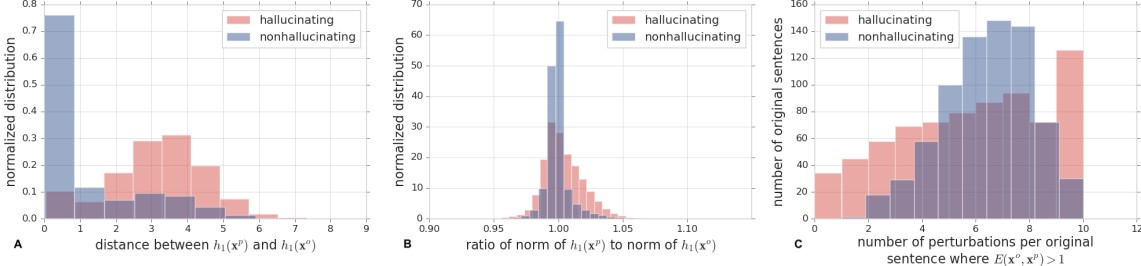

Figure 6: We compared the hidden states of the decoder for both types of perturbed sentences, hallucinating (red) and non-hallucinating (blue), with the hidden states of the decoder of the original sentence. The total area of each histogram is normalized to 1. **A.** Distance between the hidden state of the decoder for perturbed sentence and the original sentence, $\|h_1(\mathbf{x}^p) - h_1(\mathbf{x}^p)\|$. **B.** Ratio of the norm of the hidden states of the decoder of the perturbed sentence and the original sentence, $|h_1(\mathbf{x}^p)|/|h_1(\mathbf{x}^o)|$. **C.** The number of original sentences (y-axis) that have $0 < n < 10$, (x-axis) perturbed decodes with $E(\mathbf{x}^o, \mathbf{x}^p) > 1$ (see text).

## 7 DISCUSSION

In this paper we uncovered and studied a hallucination-like phenomenon whereby adding a single additional token into the input sequence causes complete mistranslation. We showed that hallucinations are common in the NMT architecture we examined, as well as in its variants. We note that hallucinations appear to be model specific. We showed that the attention matrices associated with hallucinations were statistically different on average than those associated with input sentences that could not be perturbed. Finally we proposed a few methods to reduce the occurrence of hallucinations.

Our model has two differences from production systems. For practical reasons we studied a small model and used a limited amount of training data. Given these differences it is likely that our model shows more hallucinations than a quality production model. However, given news reports of strange translations in popular public translation systems (Christian, 2018), the dynamical nature of the phenomenon, the fact that input datasets are noisy and finite, and that our most effective technique for preventing hallucinations is a data augmentation technique that requires knowledge of hallucinations, it would be surprising to discover that hallucinations did not occur in production systems.

While it is not entirely clear what should happen when a perturbing input token is added to an input source sequence, it seems clear that having an utterly incorrect translation is not desirable. This phenomenon appeared to us like a dynamical problem. Here are two speculative hypotheses: perhaps a small problem in the decoder is amplified via iteration into a much larger problem. Alternatively, perhaps the perturbing token places the decoder state in a poorly trained part of state space, the dynamics jump around wildly for while until an essentially random well-trodden stable trajectory is found, producing the remaining intelligible sentence fragment.

Many of our results can be interpreted from the vantage of dynamical systems as well. For example, we note that the NMT networks using CFN recurrent modules were highly susceptible to perturbations in our experiments. This result highlights the difficulty of understanding or fixing problems in recurrent networks. Because the CFN is embedded in a larger graph that contains an auto-regressive loop, there is no guarantee that the chaos-free property of the CFN will transfer to the larger graph. The techniques we used to reduce hallucinations can also be interpreted as dynamical regularization. For example, L2 weight decay is often discussed in the context of generalization. However, for RNNs L2 regularization can also be thought of as dynamically conditioning a network to be more stable. L2 regularization of input embeddings likely means that rare tokens will have optimization pressure to reduce the norm of those embeddings. Thus, when rare tokens are inserted into an input token sequence, the effects may be reduced. Even the data augmentation technique appears to have stability effects, as Appendix 10 shows the overall stability exponents are reduced when data augmentation is used.

Given our experimental results, do we have any recommendations for those that engineer and maintain production NMT systems? Production models should be tested for hallucinations, and when possible, the attention matrices and hidden states of the decoder should be monitored. Our results on reducing hallucinations suggest that standard regularization techniques such as Dropout and L2 weight decay on the embeddings are important. Further, data augmentation seems critical and we recommend inserting randomly chosen perturbative tokens in the input sentence as a part of the standard training regime (while monitoring that the BLEU score does not fall). We note a downside of data augmentation is that, to some extent, it requires knowing the types of the pathological phenomenon one desires to train against.

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

## 8 APPENDIX

### 8.1 NMT DECODER SCHEMATIC

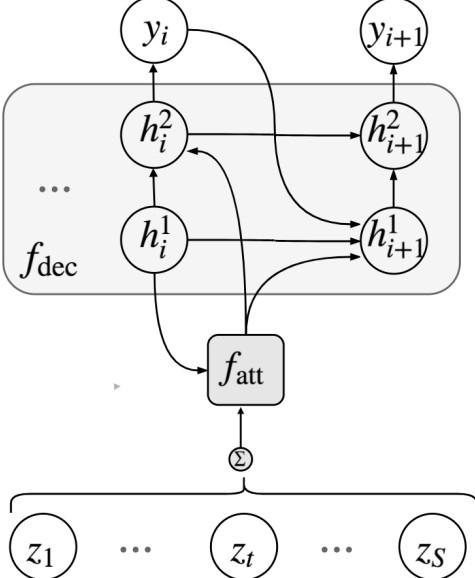

Figure 7: Schematic of the NMT decoder. The input sequence, $\mathbf{x}_{1:S}$, is encoded by a bidirectional encoder (not shown) into a sequence of encodings, $\mathbf{z}_{1:S}$. The attention network, $f_{att}$, computes a weighted sum of these encodings (computed weights not shown), based on conditioning information from $\mathbf{h}$ and provides the weighted encoding to the 2-layer decoder, $f_{dec}$, as indicated by the arrows. The decoder proceeds forward in time producing the translation one step at a time. As the decoder proceeds forward, it interacts with both the attention network and also receives as input the decoded output symbol from the previous time step.

### 8.2 ADJUSTED BLEU SCORE

Examples of pairs of sentences with different adjusted BLEU scores are as follows:

```
BLEU: 0.5
Sent 1: The role you play when creating the news is very important.
Sent 2: The part you play in making the news is very important.

BLEU: 0.09:
Sent 1: At the moment, men are overweighed by men.
Sent 2: Currently the majority of staff are men.

BLEU: 0.05:
Sent 1: Austria has also introduced controls to its southern and eastern boundaries.
Sent 2: The German government has also established the most recent and national
and international government institutions.

BLEU: 0.01:
Sent 1: " But then, suddenly, our small Jumbo was there ", says Brigitte Probst.
Sent 2: Scientists, who are not in the right place, are not in the right place.

BLEU: 0.001:
Sent 1: Mr Corbyn said that Angela Eagle, the shadow of Economic Minister, will be
represented in the economic minister, when he does not participate-for example, when
Mr. Cameron is on the
Sent 2: Mr. Mr. Mr. Mr. Mr. Mr. Mr. Mr. Mr. Mr. Mr. Mr. Mr. Mr. Mr. Mr. Mr. Mr.
Mr. Mr. Mr. Mr. Mr. Mr. Mr. Mr. Mr. Mr. Mr. Mr. Mr. Mr. Mr. Mr. Mr. Mr. Mr. Mr.
Mr. Mr. Mr. Mr. Mr. Mr. Mr. Mr. Mr. Mr. Mr.
```

As seen above, an adjusted BLEU score of $< 0.01$ means the two sentences have very few words in common.

## 8.3 ADDITIONAL EXAMPLES

Additional examples of perturbed input sentences. For many sentences, there were more than one combination of token and position that caused a hallucination. In those cases, we give several examples of different perturbations and their hallucinations.

'@@' indicates a subword token. For example, perturbing the beginning of the sentence "Vor elf..." with g@@ would give: "gVor elf..."

**Source**: Vor elf Jahren sitzt Sufjan Stevens auf der Bühne im Kölner Prime Club (heute: Luxor).
**Reference**: Eleven years ago, Sufjan Stevens sits on the stage in the Prime Club (now Luxor) in Cologne.
**NMT Translation**: About eleven years ago Sufjan Stevens sits on the stage in Cologne Prime Club (today: Luxor).
*Beginning: g@@*: EPEFA is the first year of the year.
*Beginning: P@@*: It is possible to take a look at the location of the hotel.
*Randomly: P@@*: It is possible to take a look at the location of the hotel.

**Source**: Die Leichen von 29 Toten werden von Muslimen bei den ersten Begräbnissen in Mekka durch die Straßen getragen
**Reference**: Bodies of 29 of the dead carried through the streets by Muslims as the first funerals take place in Mecca
**NMT Translation**: The bodies of 29 dead are carried by Muslims at the first slopes in Mecca by the streets.
*Beginning: er@@*: I didn 't hesitate to hesitate to contact me.
*Beginning: d*: EEEEEEEEEEEEEEEEEEEEEEEEEEEEEEEEEEEEEEEEEEEEEEEEEEEEEEEEEEEEE

**Source**: Caldiero sprach mit E! Nachrichten nach dem hart erkämpften Sieg, noch immer unter dem Schock über den Gewinn des Großen Preises von 1 Million $.
**Reference**: Caldiero spoke with E! News after the hard-fought victory, still in shock about winning the $1 million grand prize.
**NMT Translation**: Caldiero spoke with E, after the hard won victory, still under the shock of the winning of the Grand Prix of 1 million $.
*Beginning: mit*: It was said to have a lot of fun in the world.
*Beginning: werden*: Don't hesitate to contact us, if you want to be able to pay for you.
*Beginning: dass*: I 'm looking forward to having a lot of money.
*Beginning: wird*: I 'm looking forward to hearing from you.
*Beginning: (*: I 'm sure to say that I 'm looking for a lot of fun.
*Beginning: )*: I 'm looking forward to hearing from the band.
*Beginning: ich*: Thus, we are looking forward to hearing from you.
*Beginning: t*: If you 're looking for a woman, you 'll be able to get a lot of money.
*Beginning: :*: Everyone who has been able to do so, is to say that it is not a good thing to do.
*Beginning: hat*: Everyone who has been able to do so, is the most important thing to do.
*Beginning: wie*: Two years ago, he was born in the 19th century, and he was born in the 19th century.
*Beginning: einen*: I 'm looking forward to having a lot of money.
*Beginning: Ich*: Don't hesitate to contact us, if you have any questions or questions you may have.
*Beginning: A@@*: If you are looking for a lot of fun, you 'll be able to find out more about it.
*Beginning: d@@*: AVI said, " I 'm looking forward to having a lot of money.
*Beginning: Ver@@*: Looking for a lot of fun, we have a lot of fun in the world.
*Beginning: nur*: I 'm looking forward to having a lot of work.
*Beginning: uns*: Beautiful Williams, which is the most important thing to do.
*Beginning: durch*: FAQUEVE THE WILL WITH THE WILL WITH THE WITH THE WITH THE WITH THE WITH
*Beginning: kann*: Don't hesitate to contact us, but you will be able to find out more.
*Beginning: P@@*: Click here to see how you 're going to do this.
*Beginning: Der*: This is the reason why I have to say that it is not the case that it is possible to have a look at the fact that it is not possible to find out more.
*Second to end: können*: I am looking forward to hearing from you.
*Second to end: o*: We are looking forward to hearing from you.
*Second to end: de*: We are looking forward to hearing from you.
*Randomly: auf*: After the day, we have a lot of fun in the world.
*Randomly: im*: Surrounded by a lot of people, we have a lot of fun.

**Source**: Wer an einer posttraumatischen Belastungsstörung erkrankt ist, leidet unter anderem an sogenannten Flashbacks mit Atemnot, Schwindel, Herzrasen und Todesängsten.
**Reference**: Those who suffer from post-traumatic stress disorder suffer among other things from so-called flashbacks, difficulties in breathing, dizziness, palpitations and fear of death.
**NMT Translation**: Those who are suffering from a post-traumatic stress, suffer from the so-called flashbacks with breathing, darkness, heart, and death.
*Beginning: mit*: It is one of the oldest in the world.
*Beginning: Sie*: It is one of the oldest in the world.
*Beginning: werden*: Don't hesitate to contact us at any time.
*Beginning: dass*: For example, you will be able to find the right place for you.
*Beginning: sind*: Amongst other things, you will be able to enjoy a breathtaking view of the city.
*Beginning: wird*: There are also some of the world's most important tourist attractions.
*Beginning: )*: It is one of the oldest in the world.
*Beginning: als*: The hotel is located in the heart of the city.
*Beginning: e@@*: It is also an ideal place for you.
*Beginning: Ver@@*: Amongst other things, you will be able to find the right place for you.
*Beginning: nur*: OSA is one of the most popular in the world.
*Beginning:!*: For example, it is also possible to find the right place for your stay.

**Source**: Laut den Medienberichten sei jedenfalls in der Früh ein erster Bus mit rund 50 Flüchtlingen in der serbischen Grenzstadt Sid angekommen, im Lauf des Tages würden weitere Busse erwartet.
**Reference**: According to media reports, at least one bus with around 50 refugees arrived in the Serbian border town of Sid in the early hours, and in the course of the day more buses are expected.
**NMT Translation**: According to the media report, at least in the morning, a first bus with about 50 refugees arrived in the Serb border town of Sid would expect more buses.
*Beginning: werden*: Don't hesitate to contact us at any time of your visit.
*Beginning: wir*: If you are looking for a busy day, then you 'll find the most important thing to do.
*Beginning: Ich*: Don't hesitate to contact us at any time of your visit.
*Beginning: Wir*: d Laut den Medienberfüre sei jederzeit einen ersten Bus einem ersten Bus mit rund 50 Rechtlingen im Serbischen Grenzstadt Sid, im Lauf der Tage would der weiteren Busse a
*Beginning: kann*: Don't hesitate to contact us at any time.

**Source:** Neue Verhandlungen mit den Piloten
**Reference:** New negotiations with pilots
**NMT Translation:** New negotiations with the pilots
*Beginning: Die*: dass Neue News bis die Piloten
*Beginning: ich*:
*Beginning: "*: en Neue Negotiations mit den Piloten d
*Beginning: einen*: en Neue Negotiations mit den Piloten
*Beginning: Das*: de Neue News mit den Piloten
*Beginning: Wir*: d Neue Neue negotiations mit den Piloten d
*Beginning: kann*: de Neue Neue negotiations mit den Piloten die Piloten
*Beginning: c@@*: durch Neue Neue negotiations mit den Piloten
*Randomly: mit*: Neuist e mehr Jahren mit der Piloten d
*Randomly: kann*: de Neue Neue negotiations mit den Piloten die Piloten

**Source:** Für die Fed-Repräsentanten beeinflussen die Marktturbulenzen die komplexe Kalkulation, wann man die Zinsen erhöhen solle.
**Reference:** For Fed policymakers, the market turmoil adds to the complex calculus of when to raise the interest rate.
**NMT Translation:** For the Fed-Representatives, the market turbos influence the complex calculation, when we increase the interest rate.
*Randomly: uns*: For FF.C.

**Source:** Schon länger war bekannt, dass es unter der Eisdecke von Enceladus flüssiges Wasser geben muss.
**Reference:** It has long been known that there must be liquid water under Enceladus 's crust of ice.
**NMT Translation:** It was already well known that there must be fluid water under the ice cup of Enceladus.
*Beginning: Ich*: I am looking forward to hearing from you.

**Source:** Freundschaft schließen durch Backen.
**Reference:** Make friends through baking.
**NMT Translation:** Friendship by baking by baking.
*Randomly: ich*: Should you want to join us?

**Source:** In der Gesellschaft hat sich – auch wegen des Fachkräftemangels – mehrheitlich die Sichtweise durchgesetzt, dass Flüchtlingen schneller der Zugang zum Arbeitsmarkt ermöglicht werden sollte.
**Reference:** As far as the majority of society is now concerned, particularly in view of skills shortage, refugees should be allowed to gain faster access to the job market.
**NMT Translation:** Most of the time in the company – even because of the lack of expert labour – a majority of the view that refugees should be made faster from access to the labour market.
*Beginning:!*: Unfortunately, it is not possible to have a look at the situation.

**Source:** Monatelang war hinter verschlossenen Türen verhandelt, gegrübelt, debattiert und gezeichnet worden.
**Reference:** Plans have been negotiated, mulled over, debated, and plotted behind closed doors for months.
**NMT Translation:** Monatelang has been negotiating, debated and drawn behind closed doors.
*Beginning: un@@*: In the month of the month of the month of the month of the month of the month, it was a matter of course.
*Beginning: uns*: During the month of the month of the month, it was a matter of course that it is not possible for us to take a look.
*Randomly: a@@*: During the month of the month of the month, it was not possible to have the opportunity to take a look at the situation.
*Randomly: o@@*: During the month of the month of the month of the month, it was a matter of course that it was not possible to have a look at the situation.

# 9 ADDITIONAL ATTENTION MATRICES

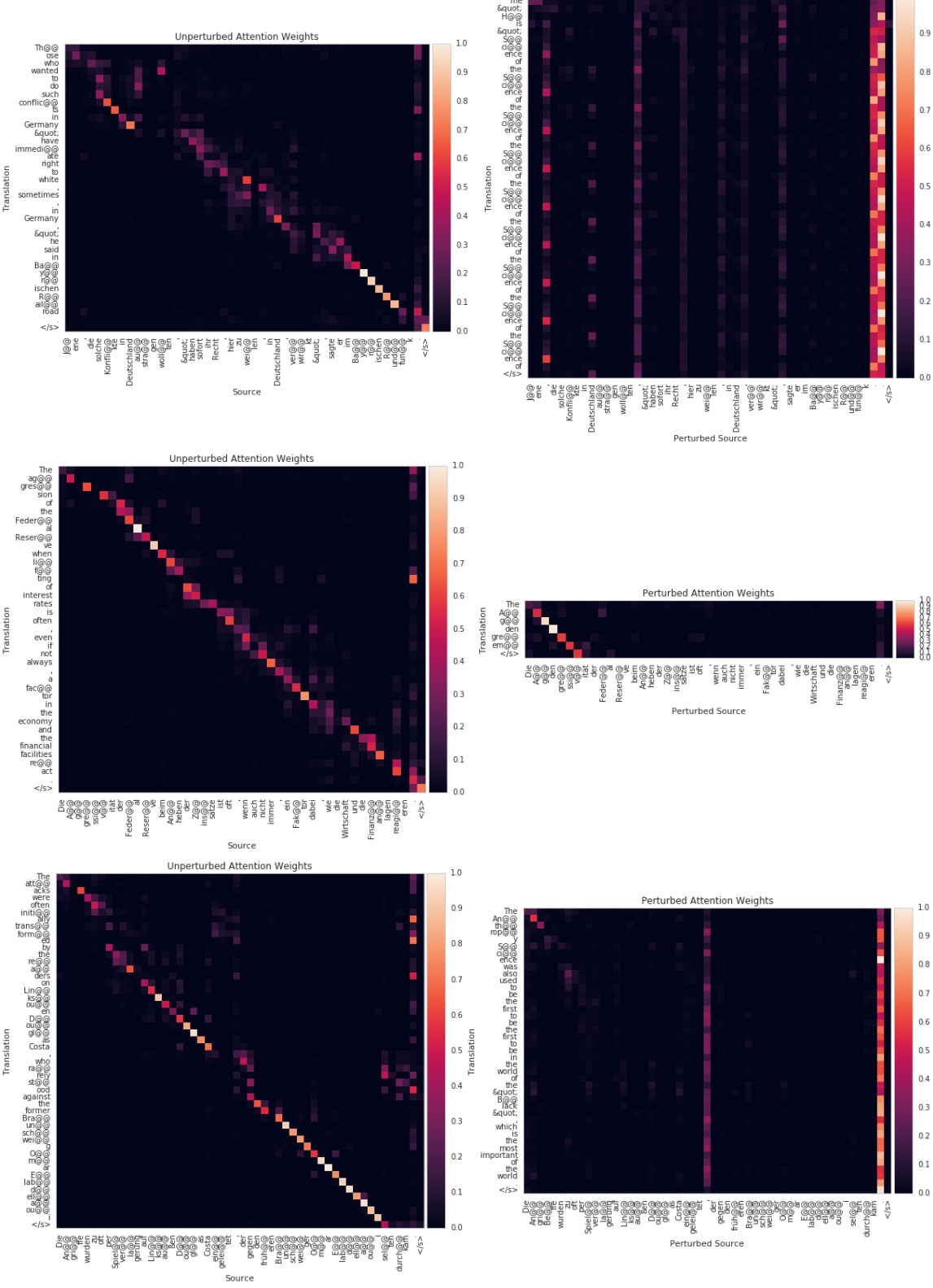

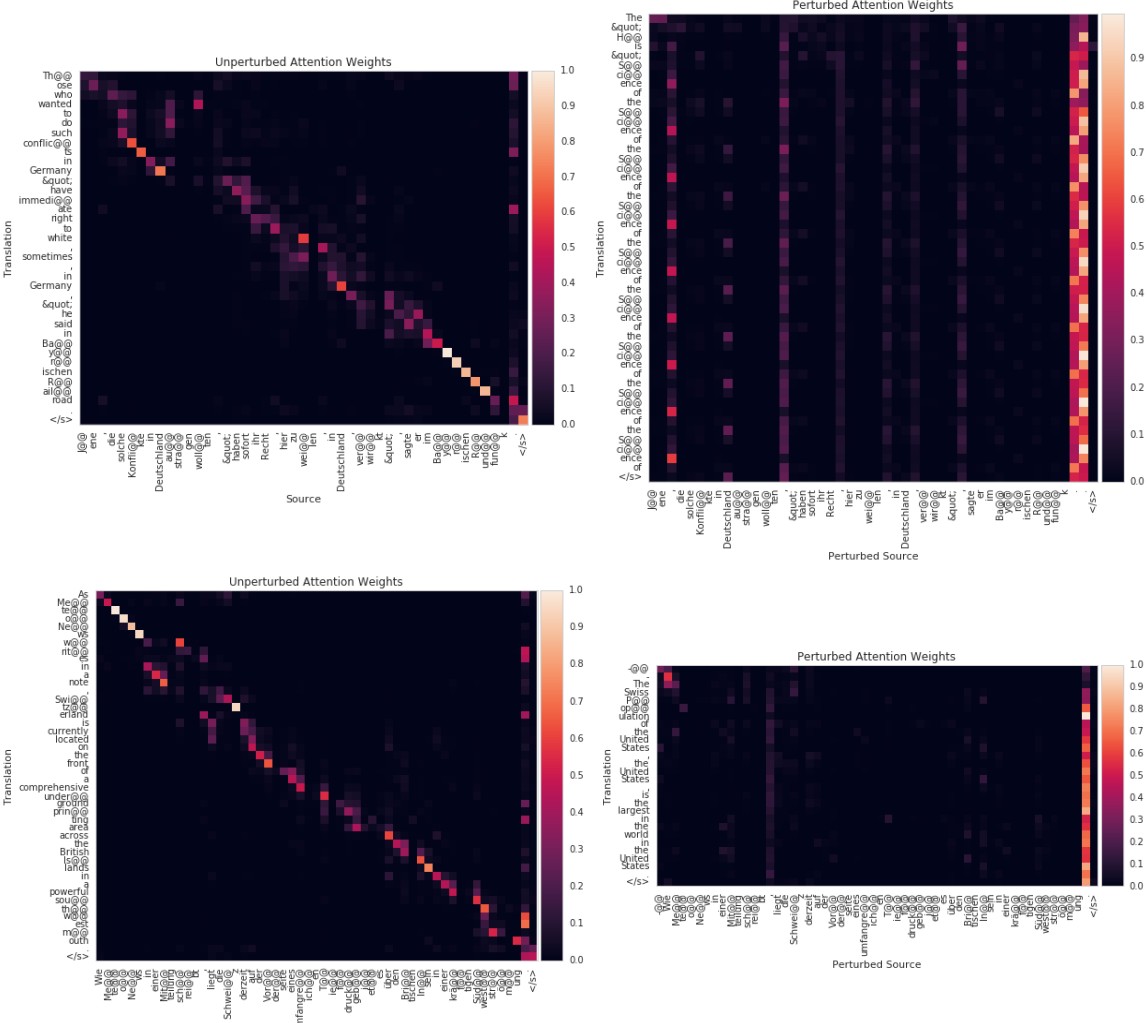

Figure 8: Example attention matrices. **(left)** Attention matrix for the original input sentence. **(right)** Attention matrix for the perturbed input sentence. All translations of the perturbed input sentence shown here are hallucinations. All decoding was done with the canonical model.

## 10 STABILITY ANALYSIS OF THE DECODER

We defined a spectrum of stability exponents for the decoder and compared them between normal translations and hallucinations (Figure 9). Concretely, we studied the stability of the decoder as a function of a given input token sequence, $\mathbf{x}_{1:S}$ of length $S$ (denoted $\mathbf{x}$ below). The sequence $\mathbf{x}_{1:S}$ is run through the encoder, whose output is processed by the attention network, finally delivering an input to the decoder. For a given input token sequence, the decoder runs until it produces an end-of-sequence token, resulting in an output token sequence $\mathbf{y}_{1:T}$ of length $T$ (or reaches a maximal decoded sequence length $T > 3S$). We were interested in studying $\frac{\partial h_T}{\partial h_0} = \frac{\partial h_T}{\partial h_{T-1}} \cdots \frac{\partial h_1}{\partial h_0}$ as many stability properties can be deduced from it. We note that if one is interested in studying $\frac{\partial h_t}{\partial x_s}$, the iterative process described by $\frac{\partial h_T}{\partial h_0}$ would still be critical to understand due to the chain rule.

We defined our spectrum of stability exponents, in analogy with Lyapunov exponents, but adapted for finite time by studying a finite-time version of Oseledets matrix, typically used in the study of chaotic dynamical systems. In particular, the $i^{th}$ stability exponent is defined as $\lambda_i(\mathbf{x}) = \frac{1}{2T} \log\left(\alpha_i(\mathbf{x})\right)$, where $\alpha_i(\mathbf{x})$ is the $i^{th}$ eigenvalue of the positive-semidefinite symmetric matrix $\left(\frac{\partial h_T}{\partial h_0}(\mathbf{x})\right)^T \left(\frac{\partial h_T}{\partial h_0}(\mathbf{x})\right)$ and $h(t)$ is the decoder state at time $t$ concatenated across all layers. We used auto-differentiation software to exactly compute the

Jacobian $\frac{\partial h_T}{\partial h_0}(\mathbf{x})$ so the complexities of the decoder circuitry were handled naturally (shown in Appendix, Section 8.1[4].

We show the distribution of stability exponents, comparing between all input sequences that could be made to hallucinate, and all those that could not (Figure 9). We show these for both the canonical model and the model trained with data augmentation. There are two observations. First, means of the distribution of the stability exponents for the canonical model, averaged over those sentences that could be perturbed to hallucinate, are statistically different than exponents averaged over sentences that could not be perturbed to hallucinate. Second, the distributions of the model trained with data augmentation show significantly reduced exponents in comparison to the canonical model.

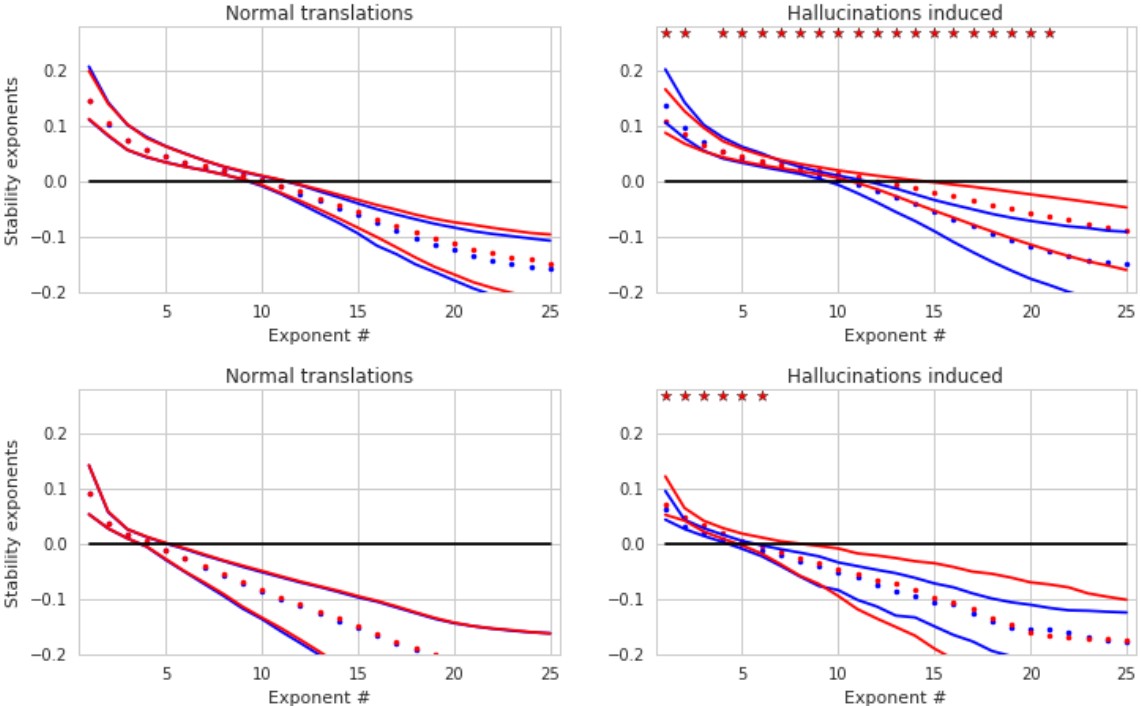

Figure 9: Stability analysis of hallucinations vs. normal translations. **(top left)** Distribution of stability exponents for canonical model for unperturbed (blue) and perturbed, no hallucination (red) input sequences that resulted in a normal translation (Median (dots) and 25% and 75% quartiles (solid lines) shown). **(top right)** Same, but translations that could be perturbed to hallucinate (blue - unperturbed, red - hallucination). Red stars denote statistical significance (U test, $p < 0.001$) in the difference in stability exponents between perturbed and unperturbed compared between normal translations and hallucinations. **(bottom left, right)** Same as top, except for model trained with data augmentation.

---

[4]Methodologically, we note that it is typical to compute Lyapunov exponents using the algorithm of (Benettin et al., 1980). This is because studying the long-time behavior of dynamical systems typically requires many thousands of system iterations. We studied the numerical stability of our direct approach to computing our stability exponents using random chaotic RNNs (Sompolinsky et al., 1988) and found that the direct approach was more than adequate for short sequences and reasonable condition numbers, so we stayed with the simpler, direct approach.

