# OpenReview forum: "Hallucinations in Neural Machine Translation"
_ICLR.cc/2019/Conference_

### Official Review · AnonReviewer3 · 2018-11-01
**Adversarial examples in NMT**

**Rating:** 7
**Confidence:** 4

**Review:**

The authors introduce hallucinations in NMT and propose some algorithms to avoid them.
The paper is clear (except section 6.2, which could have been more clearly described) and the work is original.
The paper points out hallucination problems in NMT which looks like adversarial examples in the paper "Explaining and Harnessing Adversarial Examples". So, the authors might want to compare the perturbed sources to the adversarial examples.
If analysis is provided for each hallucination patten, that would be better.

---

> ### Author Response · Authors · 2018-11-25
> **Thank you for your feedback.**
>
> Thank you for your feedback! We're glad you find this exploration interesting.
>
> We've given some thought to how hallucinations compare to adversarial examples. Like adversarial examples, hallucinations illustrate a form of instability in models which can be useful to understand why the model behaves a particular way and help propose ideas for improving stability and generalizability of models. One difference is that we aren't looking for worst case perturbations (or to perturb an input to a particular result), nor do we use gradient based methods. We show it is simple to find a perturbation that causes such a divergent hallucination. So we have similar motivations, but go about it in different ways.

---

### Official Review · AnonReviewer2 · 2018-11-03
**about the models**

**Rating:** 4
**Confidence:** 4

**Review:**

	My major concern about the work is that the studied model is quite weak.
	"All models we present are well trained with a BLEU score of at least 20.0 on the test set, a reasonable score for 2-layer models with 256 hidden units."
	"We then used the WMT De!En 2016 test set (2,999 examples) to compute the hallucination percentage for each model."
	I checked the WMT official website http://matrix.statmt.org/matrix. It shows that the best result was a BLEU score of 40.2, which was obtained at 2016. The models used in this work are about 20.0, which are much less than the WMT results reported two years ago. Note that neural machine translation has made remarkable progress in recent two years, not to mention that production systems like Google translator perform much better than research systems. Therefore, the discoveries reported in this work are questionable. I strongly suggest the authors to conduct the studies base on the latest NMT architecture, i.e., Transformer.

	Furthermore, I checked the examples given in  introduction in Google translator and found no hallucination. So I'm not sure whether such hallucinations are really critical to today's NMT systems. I'd like to see that the study on some production translation systems, e.g., applying Algo 1 to Google translator and check its outputs, which can better motivate this work.

	For the analysis in Section 6.1, if attention is the root cause of hallucinations, some existing methods should have already address this issue. Can you check whether the model trained by the following work still suffers from hallucinations?
Modeling Coverage for Neural Machine Translation, ACL 16.

---

> ### Author Response · Authors · 2018-11-25
> **Thank you for your feedback. We've run additional experiments to address your questions. pt 1.**
>
> Thank you for your feedback. As per your suggestions, we added the following additional models and experiments, resulting in the following changes.
> * The BLEU scores we reported in the paper are with greedy decoding. Since the NMT community frequently reports BLEU with beam search, we have updated our paper to reflect this. Our canonical model achieves a competitive BLEU score of 25.6 on newstest16 (https://github.com/tensorflow/nmt#wmt-german-english). We now report this in the paper.
> * We added a Transformer model to our results.
> * We perturbed the Transformer model to hallucinate and found that it hallucinates on average (over ten random seeds) 16.6% of the time (there exists a token such that 16.6% of source sentences can be made to a hallucinate). We expand on and give a discussion of the Transformer model we used in the paper.
> * You are right that coverage would be an interesting model variant to look at. We ran a coverage study and found that coverage with beam search hallucinates on average 49.4% of the time, whereas beam search hallucinates 48.2% of the time and greedy decoding hallucinates 73.3% of the time. We expand more on why we chose this version of coverage below.
> * Finally, we correlated BLEU score to perturbation percentages and did not see a decrease in perturbation percentage as BLEU score increased.  We have added an additional figure to show this. The data shows that there is a correlation coefficient of 0.33 between BLEU score and hallucination percentage. This correlation should be interpreted cautiously because we haven’t exhaustively explored the full space of models.
>
> Our paper is an analysis paper. Our goals are to quantify the phenomenon of hallucinations and explore what this tells us about training and using NMT models. To make these goals technically feasible, we extract the core NMT neural network model from the layers of techniques and fail-safes in production systems and SoTA-level competition entries. To make these goals technically tractable, we scale down the bare model which allows us to study as many hyperparameters, architectural variants, and random seeds over the thousand+ models we studied. Results we find on small models are not irrelevant. Our canonical models train to an average BLEU score of 25.6, competitive for its size, and we show that an increase in BLEU score does not correlate to a decrease in the percentage of hallucinations. In the next comment, we'll expand further on our decisions.

---

> > ### Author Response · Authors · 2018-11-25
> > **pt 2**
> >
> > It is difficult to perform exactly algorithm 1 on translation systems like Google Translate. Our analysis requires knowing the vocabulary the model used during training, but production systems are typically trained on datasets that aren't publicly available. We invite researchers who train and serve production systems to test their systems with our methodology.
> >
> > To explain further why we study a smaller neural network module in insolation, we first agree that production systems output better translations than research systems. Competition submissions also output better translations than research systems. However, our goal is to analyze and quantify a phenomenon we observed. Successful translation products deploy additional safeguards to reduce malformed outputs that are sometimes part of the model (as described above) and sometimes software, including overwriting and fixing outputs that have bad-publicity or are generally malicious. For Google Translate in particular, many examples have been logged/blogged (eg. https://motherboard.vice.com/en_us/article/j5npeg/why-is-google-translate-spitting-out-sinister-religious-prophecies and https://twitter.com/hashtag/neuralempty?src=hash We cite the former in our paper). Competition submissions also employ auxiliary techniques on top of the base model which complicates training and decoding in ways that are not well understood. For example, why does back-translation help so much? What does it change in the trained model? Thus, to hope to study the phenomenon, we remove these additional techniques from the bare NMT model. Our paper quantifies this study and documents our attempts to reduce and understand it. We believe that today's NMT systems, at the core of translation products and competition submissions are prone to hallucinations.
> >
> > Studying smaller models allowed us to tractably study many variants of our canonical model. For the size we investigated, the models used in our experiments achieve a competitive, average BLEU score of 25.6 (we previously reported the greedy BLEU score and not beam search, which the community typically reports). An RNN based NMT model (with 4x larger vocabulary and 4x bigger dimensionality compared to our models) is expected to reach 28 BLEU score ball-park as indicated here (https://github.com/tensorflow/nmt#wmt-german-english, on which our implementations are based) on the particular test set (newstest16) we’ve used. Since the WMT challenge does not require a single model, all systems with a BLEU score of 30+ incorporate additional techniques on top of the bare NMT architecture. For instance, 2016 WMT German-English winning system with a 38.2 BLEU score, uses back-translation (a data augmentation technique for MT), model ensembles, and rescores with massive Language Models on top of large Neural Networks. These additional techniques would have made studying hallucinations overly complex and masks attempts to tease out root causes of this phenomenon.
> >
> > Smaller models also allowed us to explore a dynamical systems perspective. We were unable to compute the Jacobian of the hidden states of the decoder on a larger model because it simply could not fit in memory. At this size, we can feasibly compute the Jacobian, dh(t)/dh(s), but it still takes around 20 minutes per Jacobian we wish to compute. Even after parallelization, computing Jacobians for at least 2999 sentences pushes us to the edge of reasonable scientific exploration. With a model larger in any dimension, we would not have been able to do this analysis.
> >
> > Your suggestion to study coverage is interesting. While the coverage method proposed by Tu et al. ACL'16 is nontrivial to incorporate into our systems and to test, we did a separate coverage test by providing a coverage penalty during beam search decoding (as used in Google NMT) and found that they hallucinated on average 49.4%. For reference, the canonical model decoded with beam search hallucinates on average 48.2% and greedy decoding hallucinates on average 73.3%. It appears that adding coverage did not decrease hallucinations. However, adding coverage to beam search did impact the BLEU score and lowered the BLEU score from 25.6 to 22.3.
> >
> > Thank you for giving us your feedback. We hope you will consider our goals and motivations while evaluating our work.

---

> > > ### Comment · AnonReviewer2 · 2018-11-27
> > > **Transformer**
> > >
> > > I suggest to use Transformer_big configuration, which should be much much better than the tiny config.

---

> > > > ### Author Response · Authors · 2018-11-27
> > > > **Yes, we will.**
> > > >
> > > > The transformer_big are too large to feasibly run many experiments on.  However, at your request, we are currently finishing a batch of transformer_base models (each takes about a week).  We will include the results of these models in our manuscript.

---

> > ### Comment · AnonReviewer2 · 2018-11-27
> > **the goal and the results**
> >
> > 1. I think a better goal is to quantify the phenomenon of hallucinations of STOA NMT models or systems, instead of a very simple (and maybe out-of-date) model. Even for the updated number, 25.6, it is still far more less than the best results (40.2) two years ago.
> >
> > 2. What are the configurations and the BLEU numbers of Transformer model?

---

### Official Review · AnonReviewer1 · 2018-11-03
**interesting analysis**

**Rating:** 6
**Confidence:** 5

**Review:**

I think this paper conducts several interesting analysis about MT hallucinations and also proposes several different ways of reducing this effect. My questions are as follows:

* I am very curious about how do you decide the chosen noisy words. I am also wondering what is the difference if you do choose different noisy words. Another thing, if the noisy words are unseen in the training set, will it be treated as "UNK"?
* Can you highlight what is changed in the upper right side of fig.4? It would be great if you include gloss in the figure as well.

---

> ### Author Response · Authors · 2018-11-25
> **Thank you for your feedback! We've made some clarifications.**
>
> Thank you! At your request, we have updated figure 4 to make it more clear what each part represents. We've also added to the caption to explain what the differences between the two attention matrices are. Below, we've expanded more on the questions you've raised.
>
> 1. We chose subword tokens (segmented with byte pair encoding) from our source language (German) vocabulary so we never have a noisy word that’s unseen in the training set. We’ve described how we develop our source, subword vocabulary in section 3 at the bottom of page 3. We chose these tokens as representative of the distribution of tokens: The specific tokens we’ve chosen are based on one of four types of subword tokens: common, rare, mid-frequency, and punctuation tokens. We first sorted our vocabulary of subword tokens by frequency, then formed the following groups:
>     a. Common tokens: the 100 most common tokens
>     b. Rare tokens: the 100 least common tokens
>     c. Mid-frequency tokens: After removing common and rare tokens from our sorted vocabulary of subword tokens, we sample 100 random tokens.
>     d. Punctuation tokens: All punctuation marks that exist in the vocabulary.
>     (This selection process is described in the first paragraph of section 4.)
>     Since we use BPE encoding, which segments words into sub-word units depending on their frequencies (character level co-occurrences to be precise), unseen words are never treated as UNK tokens. If a word does not appear in the training set, the BPE algorithm will segment it into the sub-words or characters that appear in our final vocabulary instead of using the UNK token.
>
> 2. Here is a further explanation of the difference in the upper right of figure 4 compared to the upper left.
> The attention matrix shows the attention weight applied to each input token in the source sentence (x-axis) as the model decodes and outputs the translated sentence (y-axis). On the upper left, we show the attention matrix of an unperturbed translation. We see weight is applied to most of the input source tokens. On the upper right, we show the attention matrix of the same source sentence, but with a perturbation at the beginning (‘und’) that causes the translation to hallucinate. We observe that weight is applied to very few input source tokens throughout translation, which is highly atypical and indicative of a broken translation.

---

### Public Comment · (anonymous) · 2018-11-18
**Flaws in your approach?**

Hi there,

When attempting to reproduce your work, we identified a couple of issues that raise some important concerns. I've enumerated our concerns below:

- Why did you use BPE tokens?

There's two issues with doing this. The first is that this does not make sense from a real world perspective: if you had used a character model and inserted characters, it resembles human typos, or a world model and inserted words, it resembles errors typically seen in spoken translation systems. However, the use of BPE is somewhere in between -- without any real world basis for this task.

The second, and much more concerning issue, is the way with which you used BPE tokens. It appears, from your examples, that you do not append a BPE token as a separate word (e.g., 'und' and not 'und@@') which would lead the model to treat the added word as part of the next/surrounding words. This is a big issue and significantly weakens the implications of your analysis. We've tried to reproduce your work with a word level model and have failed (we think it's most likely because of the second issue with how you inserted BPE tokens).

- Why is your baseline so weak?

You need a stronger baseline. Furthermore, you need to present results on the clean dataset with your baseline and with your improved models. It's meaningless to have a model which has fewer hallucinations, if it means significantly degraded performance on the original dataset.

- Your algorithm is very brute-forcey.

The fact that you define your hallucination accuracy to be, whether one of 100 words (in various positions) caused a hallucination, overstates the significance of the hallucination issue. It could, very well be, that whichever perturbation ends up causing a mis-translation has, in fact, significantly altered the meaning of the sentence.

---

> ### Author Response · Authors · 2018-11-26
> **Thank you for your feedback. pt 1.**
>
> Thank you for your interest and feedback. With this paper, we introduce and document a novel phenomenon. We hope the community will do as you have done and explore stability, hallucinations, and the internals of their models.
>
> Respectfully, we strongly reject the implication that our methodology is flawed and are confident in our findings. There may be any number of reasons why your results differ from ours, from minor bugs to simple hyper parameter differences. We find the phenomenon of hallucinations to be robust over all our experiments, which include over a thousand models with many hyperparameter settings, random seeds, and architectural variants.
>
> Below, we answer each question:
>
> - Why did you use BPE tokens?
>
> We trained all our models with BPE/WPM because it is the common standard in NMT research and production [Google NMT (Wu et al. 2016) uses word-piece model, Transformer (Vasvani et al. 2017) and its derivatives for WMT-17, 18 are using byte-pair encoding (Bojar et al. 2017-18).]. We chose to also use BPE for perturbations to stay consistent with the model (we don’t re-tokenize the sentence after perturbing it), and allows us to test a mixture of word and character perturbations. That being said, our methodology is much closer to perturbing models with full words as the majority of tokens we perturb with are either full words, punctuation or single characters. Of the tokens we chose as perturbing tokens, 75% of all common tokens, and 37% of all rare tokens are full words. The vast majority of rare tokens (~80%) are single Chinese, Korean, or Arabic characters. In our text, we give examples of realistic perturbations with both full words, for instance inserting “und,” the German word for “and” (taken from figure 5 (attention matrix), and punctuation. Here are two examples:
> Original input: In der medizinisch-behandelten Gruppe löste sich im Vergleich dazu der Diabetes vollständig nur bei einem Prozent und teilweise bei nur etwa zwei Prozent.
> Original translation: In the medical and treatment group , the diabetes solved the total only at a per cent and in some cases only two per cent of the diabetes solved .
> Reference: In the medically-treated group , by comparison , diabetes resolved completely in only 1 percent and partially in only about 2 percent .
> Perturbed input: und In der medizinisch-behandelten Gruppe löste sich im Vergleich dazu der Diabetes vollständig nur bei einem Prozent und teilweise bei nur etwa zwei Prozent .
> Translated perturbed: The company has been able to provide a new and more efficient solution for the company .
>
> Original input: Gauselmann wünscht sich , dass die Mitgliedschaft im Schachclub und auch freundschaftliche Kontakt zum Tennisclub &quot; Rot-Weiß &quot; als Ausdruck seiner Verbundenheit mit der Kurstadt gesehen wird .
> Original translation: Gauselmann wants to see that membership in the chess club and also friendly contact with the tennis club &quot; Rot-Weiß &quot; is seen as an expression of his commitment to the city city .
> Reference: Gauselmann wants his membership of the chess club as well as his friendly contact with the &quot; Red-white &quot; tennis club to be seen as an expression of his ties with the spa town .
> Perturbed input: . Gauselmann wünscht sich , dass die Mitgliedschaft im Schachclub und auch freundschaftliche Kontakt zum Tennisclub &quot; Rot-Weiß &quot; als Ausdruck seiner Verbundenheit mit der Kurstadt gesehen wird .
> Translated perturbed: The Memory of the Science of the Science of the Science of the Science of the Science of the Town Square , the Cathedral , is a new and most popular place .
>
> We give further examples of in section 8.3 of the appendix.
>
>
> - Using BPE tokens.
>
> We append, prepend, replace, etc. as tokens appear in the vocabulary. The vocabulary includes words, subword tokens, and characters. "und" for example, appears as both "und" (word) and "und@@" (subword) in the vocabulary. We're sorry this was misleading.
> Further, since we do not re-tokenize after adding perturbations, the NMT model will always see “und@@ Guten morgen” and never “<UNK> morgen.” In this case, whether a token is a subword or a full word token should not be more informative of how likely a sentence is to hallucinate than the stability of that particular token.

---

> > ### Author Response · Authors · 2018-11-26
> > **pt 2.**
> >
> > - Why is your baseline so weak?
> >
> > We have chosen to use small models for the sake of large scale analysis - note each data point comes from 10 separately trained NMT models. Additionally, we are interested in the recurrence of machine translation models and use a dynamical systems approach in one section of the paper. Jacobian computations are notoriously computationally expensive, and it would be technically infeasible to compute Jacobians on larger models (we would run out of memory). Even still, a single Jacobian computation takes around 20 minutes on CPU with Autograd. To compute the Jacobian for a larger model for 2999 sentences, even after parallelizing, would make scientific exploration infeasible. By keeping our models small, we can test over a thousand of models and explore more hypotheses. That being said, for the size of our models, our baseline BLEU score (25.6 with beam search) is reasonable as models above 30 BLEU very quickly become complicated due to the use of hard-to-analyze techniques such as model ensembling, etc.
> >
> > We aren’t sure what you mean by "present results on the clean dataset with your baseline." Our baselines are using the original clean WMT En-De train, dev and test sets.
> >
> > Different NMT systems have different values. In production, one might value stability of translations and knowing when the model is hallucinating and balance that tradeoff with accuracy. We study both how to stabilize models and detect, via the attention matrix, when the model is hallucinating.
> >
> > - Your algorithm is very brute-forcey.
> >
> > Yes it is simple :-). Our perturbation pipeline is favorable because of its simplicity and ease of adaptability. Yes, our perturbation pipeline could alter the meaning of sentences. However, one would not expect a completely different translation by adding a word, such as ‘and’, and this is precisely what we document. Previous work shows reastic perturbations (like typos) (such as: https://arxiv.org/abs/1711.02173) that do not semantically alter the meaning of sentences, but the example perturbed translations given in those texts are not as drastically different as we show. Further, we have provided examples of perturbations that do not alter semantic meaning, but still result in hallucinations (for example: all punctuation added to the beginning or end of a sentence, or adding “und” (German for “and”) as shown above). We aim to provide a framework and categorize a phenomenon that can help improve robustness of translation systems through identifying and understanding where the model slips up.
> >
> > As you may have different values for the model or system you’re building, we welcome you to try other types of perturbations (as you seem to have) and even modify the criteria for what is a hallucination based on your own understanding of your models (shift the threshold, include perplexity, change the weights of the adjusted BLEU score). We provide a simple setup, and make a case for the community to explore stability, hallucinations, attention, and the decoder’s dynamics, as you have already begun to do.

---

### Meta-Review · Area_Chair1 · 2018-12-15
**sufficiently solid but not particularly exciting**

**Confidence:** 5
**Recommendation:** Reject

**Metareview:**

Strengths

-  Hallucinations are a problem for seq2seq models, esp trained on small datasets

Weankesses

- Hallucinations are known to exists, the analyses / observations are not very novel

- The considered space of hallucinations source (i.e. added noise) is fairly limited, it is not clear that these are the most natural sources of hallucination and not clear if the methods defined to combat these types would generalize to other types. E.g., I'd rather see hallucinations appearing when running NMT on some natural (albeit noisy) corpus, rather than defining the noise model manually.

-  The proposed approach is not particularly interesting, and may not be general. Alternative techniques (e.g., modeling coverage) have been proposed in the past.

-  A wider variety of language pairs, amounts of data, etc needed to validate the methods. This is an empirical paper, I would expect higher quality of evaluation.

Two reviewers argued that the baseline system is somewhat weak and the method is not very exciting.